# Detection of Deaths Caused by Hyperkalemia

**DOI:** 10.3390/biomedicines13010222

**Published:** 2025-01-17

**Authors:** Małgorzata Żulicka, Kamila Sobczak, Dominik Kowalczyk, Sylwia Sikorska, Wioletta Arendt, Marta Hałas-Wiśniewska

**Affiliations:** 1Students Research Group of Cell Biology and Ultrastructure, Department of Histology and Embryology, Faculty of Medicine, Collegium Medicum in Bydgoszcz, Nicolaus Copernicus University in Toruń, 85-092 Bydgoszcz, Poland; malgorzatazulicka@gmail.com (M.Ż.); kamilas77533@gmail.com (K.S.); domin070@wp.pl (D.K.); 2Medical Laboratory for Pediatric Oncology and Hematology, Immunopathology and Genetics Unit, Central Clinical Hospital of the Medical University of Lodz, University of Lodz, 92-216 Lodz, Poland; sikorskasylwia2@gmail.com; 3Department of Histology and Embryology, Faculty of Medicine, Collegium Medicum in Bydgoszcz, Nicolaus Copernicus University in Toruń, 85-092 Bydgoszcz, Poland; warendt@cm.umk.pl

**Keywords:** hyperkalemia, death, potassium ions, post-mortem

## Abstract

Under normal conditions, potassium is predominantly found within cells. The concentration gradient of sodium and potassium ions between intracellular and extracellular spaces enables signal transmission through membrane depolarization. The disruption of this transcellular process leads to elevated potassium ion levels in the extracellular space, and thus in the blood, a condition known as hyperkalemia. Clinically, hyperkalemia may present as cardiac arrhythmias, muscle weakness, and palpitations. The post-mortem accumulation of potassium ions in various human tissues and organs, such as the heart, liver, kidneys, lungs, and vitreous body, particularly in cases of overdose, has been an area of research interest for years. Unfortunately, deaths caused by hyperkalemia are difficult to identify due to their non-specific symptoms and are often misinterpreted as cardiovascular-related. Furthermore, most potassium ion concentration tests developed in recent years are non-specific, have limitations, or are based on outdated techniques. Consequently, alternative methods, such as histopathological tissue analysis, potassium concentration assessment in the vitreous body, and aldosterone level measurement, show promise for improving the post-mortem detection of exogenous hyperkalemia.

## 1. Introduction

Physiologically, the intracellular concentration of potassium ions is about 30 times higher than in the extracellular space. Signal transmission through membrane depolarization is possible due to the difference in sodium and potassium ion concentrations between the inside and outside of the cell [1]. An elevated potassium concentration in the blood indicates an increase in potassium levels outside the cell. This disrupts the concentration gradient between the intracellular and extracellular environments, significantly lowering the threshold potential. Additionally, sodium efflux is impaired, which hinders both stimulus formation and propagation. Hyperkalemia is defined as a condition in which serum or plasma potassium levels exceed 5.5 mEq/L. When potassium concentrations reach 8–9 mEq/L, symptoms such as skeletal muscles weakness, cardiac arrhythmias, and disturbances of consciousness may occur [2]. Moreover, hyperkalemia significantly affects the circulatory system. By reducing the threshold potential, it increases the cell membrane’s conductivity to potassium ions in cardiomyocytes, leading to a deceleration of the slow resting depolarization. As a result, action potentials are generated less frequently in the pacemaker cells of the heart, manifesting as pointed T waves, widened QRS complexes, bradycardia, and even asystole. In the later stages, this condition can progress to ventricular fibrillation and ultimately, death [1,3,4,5,6].

After death, the ion concentration gradient across cell membranes dissipates. As cells undergo lysis, their permeability increases, leading to the equalization of ion concentrations in the intracellular and extracellular environments. This effect is one reason why deaths resulting from potassium poisoning may go undetected. The non-specific symptoms of hyperkalemia make it challenging to recognize a crime, as such deaths may be classified as cardiovascular-related, the most common cause of death worldwide [7]. Estimating the scale of this phenomenon is difficult, as hyperkalemia-related symptoms and deaths typically do not raise suspicion, underscoring the need to find reliable methods to detect potassium overdose. Figure 1 illustrates the primary locations of sodium and potassium ions both ante-mortem and post-mortem.

Currently, the most common way to establish potassium overdose as a cause of death relies on circumstantial evidence, such as the surroundings of the deceased [3,6]. Detecting these deaths is essential not only to identify crimes but also to recognize potential medical errors, such as administering drugs dissolved in potassium chloride instead of sodium chloride.

Research on deaths caused by hyperkalemia is limited, with most studies lacking a statistically significant sample size. To date, there are only a few publications, consisting mainly of single-case reports and isolated detection methods. The purpose of this review is to analyze current approaches and identify potential strategies for detecting deaths caused by exogenous hyperkalemia.

## 2. Materials and Methods

This review paper was prepared based on the literature reports available in the PubMed database. The following keywords were used in the search: ‘hyperkalemia’ and ‘death’. Publications were selected based on information regarding potassium ion levels at the time of death. Papers describing hyperkalemia resulting from diseases or diet were excluded. Due to limited available material, the review includes publications from the period 1969–2024. The publication selection process is illustrated in Figure 2.

## 3. Potassium Ion Concentration in Different Body Locations

### 3.1. Potassium Ion Concentration in the Blood

It is widely accepted that potassium ion concentration in the blood is indeterminate after death [3,8]. In living organism, approximately 98% of potassium ions are found intracellularly, with a concentration of around 150 mEq/L, while its extracellular levels range between 3.8 and 5.5 mEq/L [3,5]. Post-mortem potassium ion concentrations equalize across the cell membrane due to increased permeability. As extracellular potassium ion concentration rises to approximately 25–80 mEq/L, it becomes impossible to determine whether it increased ante-mortem or after death [3]. This was confirmed by a reported case of double suicide involving potassium aspartate [9]. Both individuals showed significantly elevated post-mortem blood potassium levels, which, based on toxicological analysis and evidence found at the scene, were deemed the cause of death. The observed potassium levels reached 49.7 mEq/L in the man and 62.8 mEq/L in the woman, respectively. Both values were within the range considered typical in post-mortem blood potassium levels, highlighting the challenge pathologists face when trying to confirm or exclude unnatural causes of death. These findings underscore the importance of collecting and analyzing items near the deceased, such as the composition of drugs taken or the contents of intravenous infusions, in investigating the causes of death in individuals with high blood potassium levels [3,6].

### 3.2. Potassium Ion Concentration in Tissues

The literature reports show the possibility of precipitation of potassium chloride crystals in tissues as a result of potassium poisoning. Research by Coulibaly et al. showed that lanceolate-shaped salt deposits formed in the tissues of fetuses aborted through potassium chloride injection [10]. These deposits were observed in the endocardium, epicardium, myocardium, liver, adrenal gland, testicle, spleen, kidneys, and lungs [10]. However, it is worth noting that the study group was limited, with KCl abortions conducted on only two fetuses (at 30 and 36 weeks). Additionally, fetal tissue response may not correspond to adult tissue reaction. Although the studied fetuses were in the advanced stages of development, their findings may apply to newborns but not necessarily to adults. Furthermore, the KCl doses used in the study were significantly higher than in typical overdoses [3]. In adults, microscopically and macroscopically observable changes in tissues following potassium poisoning include pulmonary engorgement, myocardial edema, and congestion of various organs [3,8,9,10,11].

Zhang et al. investigated the use of scanning electron microscopy (SEM) and electrical spectrum microanalysis for detecting fatal potassium overdoses [11]. The authors compared skin samples from the injection site with the corresponding sample from the other untreated limb. Both samples were examined using SEM and electrical spectrum analysis on randomly selected points. The injection site skin sample showed observable damage, and analysis of the electrical spectrum revealed a significantly higher potassium ion peak compared to the undamaged skin. However, it is important to note that the deceased individual examined by Zhang et al. was also characterized with an elevated potassium ion concentration in the blood and in the vitreous body [11]. Thus, it remains unclear whether a similarly high potassium peak would be observed in cases where blood potassium levels are not as high. Furthermore, a key limitation of this method is its technical complexity, which currently prevents routine post-mortem testing for potassium ion overdose in forensic investigations [11].

In vivo tests were also conducted on rabbits to examine the effects of fatal potassium poisoning. Macroscopically, myocardial dilatation and congestion were observed, while microscopically, changes in the myocardial Z-line, enlargement of Bowman’s capsule, and apoptosis with phagocytosis of brain cells were confirmed. It is worth noting that extremely high salt concentrations were used in their study, with solutions of 1% (approximately 10,000 mg/kg) and 0.3% (approximately 3000 mg/kg) potassium chloride [12]. The lethal dose of potassium chloride is generally considered to be 30–35 mg/kg body weight. This suggests that greatly exceeding the lethal dose may lead to more pronounced physiological changes than a minimal overdose. It is also important to consider that the human body may metabolize high potassium ion concentrations at a different rate [12].

A patient described by Park et al. attempted suicide via the intravenous injection of potassium salt [13]. Due to incorrect needle insertion, the injection was administrated subcutaneously, resulting in extravascular leakage that caused third-degree chemical burns and extensive necrosis. The necrosis extended into the subcutaneous tissue and blood vessels, leading to a loss of nociception and the appearance of bluish patches on the skin around the injection site. Scabs formed as a result of burns can disrupt blood flow and reduce tissue elasticity. In such cases, the lesions are visible both macroscopically and microscopically on the skin [13].

The heart is an organ especially prone to hyperkalemia-related damage due to its dependence on the sodium–potassium concentration gradient. Disturbances in the membrane potential may manifest as hypotension, bradycardia, and even asystole. Consequently, deaths caused by hyperkalemia may be misinterpreted as cardiovascular-related [14].

Wetherton et al. described four cases of fatal potassium injection [15]. In the first two cases, hyperkalemia’s severe impact on the heart muscle initially led to the misclassification of death as being due to cardiovascular causes [15]. The cardiac effects of potassium overdose have been confirmed in numerous other studies. John et al. reported cases of severe cardiovascular changes resulting in bradycardia, while Battefort et al. observed asystole due to hyperkalemia [16,17]. Madan et al. documented arrhythmia with ventricular tachycardia in one of their patients [18]. Additionally, Iijima et al. noted tented T waves and reduced P waves with electrocardiography, and Illingworth et al. reported widened QRS complexes and tall T waves in patients, with one case progressing to cardiac arrest with asystole [19,20]. These cases suggest that hyperkalemia may cause tissue changes. However, whether these would include detectable alterations remains unknown, as microscopic examination for this purpose was not conducted [21].

The kidneys are another organ affected by hyperkalemia-induced pathologies. Although a healthy individual typically removes excess potassium ions efficiently through urine, higher concentrations may overwhelm the kidney’s adaptive capacity. Hyperkalemia can impair flow in the distal part of the nephron and stimulate aldosterone secretion [17]. Coulibaly et al. observed excessive potassium accumulation in kidneys of patients with hyperkalemia [10]. It is also possible that elevated aldosterone levels shortly before death might still be detectable post-mortem, allowing for the confirmation of potassium overdose [10].

The lungs are also affected by hyperkalemia. In the vast majority of potassium overdose cases, swelling and congestion of the organ are observed [3,9,11,22]. Although these are the most common changes noted during autopsy, their presence may suggest death due to hyperkalemia. However, it cannot be considered characteristic of this condition [23].

An oral overdose of potassium can cause irritation of the gastrointestinal mucosa [24,25]. Hyperkalemia may also lead to paralytic ileus, gastrointestinal perforation, and small intestine ulcerations [5]. In a case described by Su et al., gastrointestinal bleeding was observed [24]. These examples may suggest damage that could potentially be identified during a histopathological examination. However, this contrasts with the findings of Bertol et al., who examined potassium overdose casualties via histopathological changes but observed no abnormalities [26]. Similarly, Bhatkhande et al. explicitly questioned the usefulness of measuring tissue potassium ion concentrations. Both teams primarily focused on biochemical parameters, without specifying which histopathological lesions were assessed [22].

### 3.3. Potassium Ion Concentration in Vitreous Body

The vitreous humour is the gelatinous fluid filling the eyeball, located between the crystalline lens and the retina [27]. The idea of using potassium levels in the vitreous body to determine the time of death has been considered before, with early studies suggesting it may be influenced by body temperature [28]. Recent studies showed a linear correlation between the time since death and potassium levels in the vitreous humour, which was supported through the examination of 100 deceased patients [29]. For the first 100 h post-mortem, potassium concentration in the vitreous humour gradually increases due to the autolysis of choroid. It was also observed that the rate of potassium release was affected by the circumstances of the deceased. In cases of traumatic death, a more rapid rise in potassium levels was noted within the first 6 h after death [30].

Further evidence suggests that the circumstances of death may impact potassium levels in the vitreous humour. Elevated temperatures can accelerate electrolyte release, while drowning may cause variations in potassium level measurements [31]. This supports the hypothesis that hyperkalemia might be identified post-mortem by identifying a faster rate of potassium release into the vitreous humour compared to other causes of death, caused by a steeper concentration gradient between the choroid and vitreous humour created by elevated systemic potassium levels. A remaining challenge lies in the precision of analysis, especially in older studies with outdated diagnostic tools. A 2022 animal study demonstrated that the Element DC analyzer provides precise measurements of potassium in the vitreous humour from animal corpses, offering new possibilities for tests with greater accuracy [32]. Table 1 shows a summary of various parameters from selected studies describing post-mortem potassium ion concentrations.

## 4. Post-Mortem Detection of Hyperkalemia

Available research on hyperkalemia has primarily focused on basic biochemical analyzes rather than other aspects, such as histopathological changes. However, it is worth noting that potassium ion concentration tests are considered non-specific and have limited interpretative value [33]. This focus has led to the omission of alternative methods that might offer greater effectiveness, particularly due to improved parameter stability.

We suggest that the histopathological analysis of tissues could be a valuable area for investigation. The organs of interest include the heart muscle, kidneys, small intestine, skin, and lungs, with the kidneys being particularly susceptible to hyperkalemia [5,10]. Samples should also be examined for the presence of salt deposits [10]. Current research does not provide sufficient evidence of their occurrence in adults. Since clear histopathological indicators of potassium overdose are not available, the evaluation should include a comparison of samples from suspected hyperkalemia casualties with those from individuals who died from other causes. Additionally, as some studies were conducted in the 1980s, the methods used may now be outdated, highlighting the importance of replicating and expending this research.

The main advantage of microscopic analysis is its low technical requirements. However, its application is limited by the fact that most pathologies detected microscopically are non-specific. Some lesions may be more commonly associated with hyperkalemia, such as pulmonary, cardiac congestion, and edema. Renal lesions may be more characteristic of hyperkalemia, but further research is necessary to confirm this.

The concentration of potassium ions in the blood is a diagnostically valuable parameter only up to the level of 10 mEq/L. This poses a challenge, as potassium concentration increases rapidly after death. While blood test would be the simplest method to confirm potassium poisoning, it has limited diagnostic value. Based on the literature review, we suggest that this method may be useful only in cases of extreme potassium overdose. However, even in such cases, this test alone would not be sufficient and should be supplemented with additional analyzes.

Although it has not yet been studied extensively, measuring potassium ion concentration in the vitreous humour appears to be a promising method. Given the linear relationship between the time after death and the potassium ion concentration in the vitreous humour, any deviations from this norm may indicate potassium intoxication. This approach requires first establishing a baseline curve of potassium ion concentration changes over time in the vitreous humour among individuals who died from causes other than hyperkalemia. It would also be necessary to examine whether factors such as age influence this attribute. Some of these data may be obtained from the existing literature reports on hyperkalemia. We estimate that this could be one of the more accurate methods for detecting potassium poisoning. However, a significant drawback is the difficulty in obtaining samples, and this method would also exclude cadavers exposed to high temperatures, such as those involved in fires, as electrolytes are released more quickly in such conditions [3].

In our view, another potentially effective post-mortem method for detecting exogenous hyperkalemia could be the analysis of endolymph in the membranous labyrinth. Under physiological conditions, the endolymph contains a significantly higher potassium concentration compared to the cytoplasm of the surrounding epithelial cells. In the article from 2006, the author assesses the potassium concentration for the endolymph as 157mM in comparison to 5mM in the blood plasma. This elevated potassium level distinguishes the endolymph from other extracellular fluids [34]. This unique ionic balance means that, as a result of changes in cell membrane permeability after death, potassium ions would likely flow into the cells rather than outward, impacting the rate of concentration equalization. The dynamics of potassium ion concentration shift between intracellular and extracellular environments and may depend on the amount of potassium salts absorbed ante-mortem. Detecting abnormally high potassium levels in endolymphatic tissue post-mortem, along with a faster decline in concentration and additional evidence (such as elevated potassium levels in the blood, cerebrospinal fluid, or specific microscopic changes), may aid in identifying deaths due to hyperkalemia.

While a faster ion flow may narrow the window for accurate post-mortem detection, it could nonetheless serve as an additional criterion, suggesting hyperkalemia as the cause of death. With a higher potassium concentration gradient between the cell interior and the extracellular matrix, the equalization of ion concentrations is expected to occur more rapidly.

Post-mortem changes in ion concentrations within the endolymph and perilymph of guinea pigs have been studied. Although, the study focused on physiological changes, unrelated to hyperkalemia, they may help to establish a baseline data for subsequent analysis of hyperkalemia potassium overdose cases. Changes during the first 40 min post-mortem are attributed to ion flow through the endothelium of the endolymphatic sac along the concentration gradient, while further changes reflect the equalization of concentrations between blood, endolymph, and perilymph. In live guinea pigs, the potassium ion levels range from 114 to138 mEq/L [35]. Initial changes after death proceed rapidly, with potassium concentrations dropping from an average of 128.63 mEq/L to 122 mEq/L within 10 min and to 97 mEq/L 12 min post-mortem. Melichar et al. studied changes in intraluminal potassium concentration 20–60 min after anoxia [36]. The observed level dropped from 138 mEq/L to 130 mEq/L within the first 12 min, and after an hour, it decreased to 110 mEq/L. Both studies showed a similar trend as the potassium concentrations decline at a similar rate despite differing baseline values [36]. In the study by Rodgers et al., the average decline rate was 0.62 mEq/L per minute, compared to 0.47 mEq/L per minute in the study by Melichar et al. [35,36]. Differences in baseline concentrations may result from varying sampling times or individual variation. In hyperkalemia, the increased concentration gradient may accelerate ion flow. Currently, there are no reports on the post-mortem potassium concentration changes in the endolymph of other species, including human. Similarly, the effect of exogenous hyperkalemia on the ion balance between the endolymph and the surrounding epithelium has not been studied yet [35,36].

Research should begin by assessing how the ionic composition of the endolymph changes over time after death in humans. Following this, it should be analyzed whether potassium overdose impacts this process. For obvious reasons, conducting such studies with human subjects is challenging. Retrospective studies could be helpful, provided that endolymph samples were collected. Endolymph samples can be obtained directly from the endolymphatic duct or by exposing the endolymphatic sac on the posterior surface of the petrous part of the temporal bone [37].

Another potential method for detecting hyperkalemia post-mortem is testing aldosterone levels. In cases of hyperkalemia, aldosterone concentrations physiologically increase [38]. Currently, there are no studies examining post-mortem changes in aldosterone levels. Elevated blood aldosterone levels post-mortem could indicate potassium overdose ante-mortem. Although the pattern of post-mortem change in blood aldosterone concentration has not yet been thoroughly studied, it would be possible to calculate its concentration based on its half-life in the human body (about 20 min) [39,40]. Potential advantages of such an approach include the ease of sampling, resistance to post-mortem concentration changes, and the ability to calculate the exact concentration of aldosterone before death (based on the time elapsed since death and the half-life). This allows for the identification of elevated aldosterone levels secondary to hyperkalemia. Nonetheless, it has limitations, such as reduced reliability in individuals with endocrine disorders(for example adrenal dysfunction) and a short time to sample due to the short half-life of aldosterone in the blood. Table 2 outlines the potential methods for identifying deaths caused by exogenous hyperkalemia.

## 5. Conclusions

There is a general opinion that fatal potassium overdoses are undetectable post-mortem due to a lack of adequate forensic tools. To date, this cause of death has primarily been identified based on circumstantial evidence, such as the deceased’s surroundings, injection marks, or witness testimonies [3,6]. Additionally, the non-specific symptoms of hyperkalemia can lead to misinterpretation as cardiovascular-related deaths [15,16,17].

The topic is challenging to analyze, as most research relies on single-case studies, which are difficult to compare due to varying examination times for individual parameters that change rapidly post-mortem. Therefore, the aim of this review was to propose potential methods for detecting deaths resulting from exogenous hyperkalemia.

In summary, there is a significant lack of research on deaths due to hyperkalemia, leading to a paradox; insufficient research makes it difficult to detect these deaths, and when they go undetected, conducting further research becomes more challenging. We proposed a histopathological analysis, measuring the concentration of potassium ions in the vitreous body or endolymph as well as blood aldosterone levels as the most promising options. It is also worth mentioning the diagnostic potential of combining electrocardiographic (ECG) analysis with machine learning (ML). Currently, in this context, reports mainly concern studies on living patients. The application of ML technology in the case of death is a challenge due to physiological changes after death, such as tissue breakdown and lack of electrical activity of the heart. However, pilot studies suggest that ML can support the analysis of pathology in a forensic context, e.g., in detecting patterns indicating potassium concentration in tissues in the case of death caused by hyperkalemia. However, further research is necessary to develop procedures adapted to the specificity of post-mortem changes [41,42]. The methods outlined in this review are not without their limitations, each with varying degrees of error. However, analyzing combinations of these parameters may help to ultimately identify hyperkalemia.

When hyperkalemia is suspected as a cause of death, circumstantial evidence should also be considered. Identifying such deaths is crucial not only in forensic contexts but also in medical errors. Using potassium chloride instead of sodium chloride as a diluent can result in a fatal outcome for the patient, making error prevention vital [15,43,44].

## Figures and Tables

**Figure 1 biomedicines-13-00222-f001:**
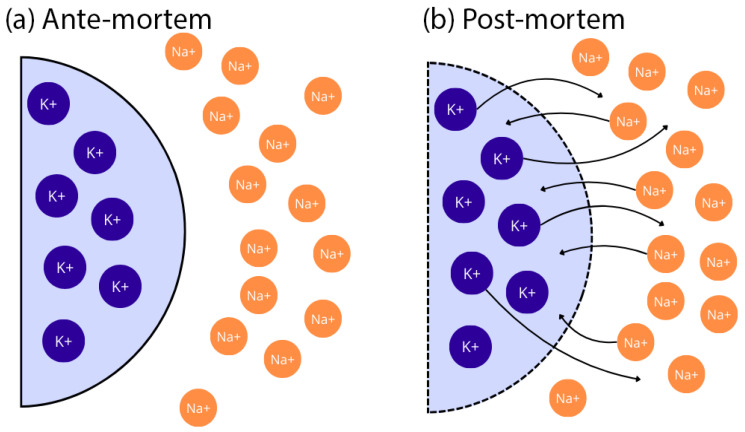
Diagram of potassium ion gradients: (**a**) ante-mortem and (**b**) post-mortem. Prepared based on [1]. Na^+^—sodium cation, K^+^—potassium cation, arrow—ion flow direction.

**Figure 2 biomedicines-13-00222-f002:**
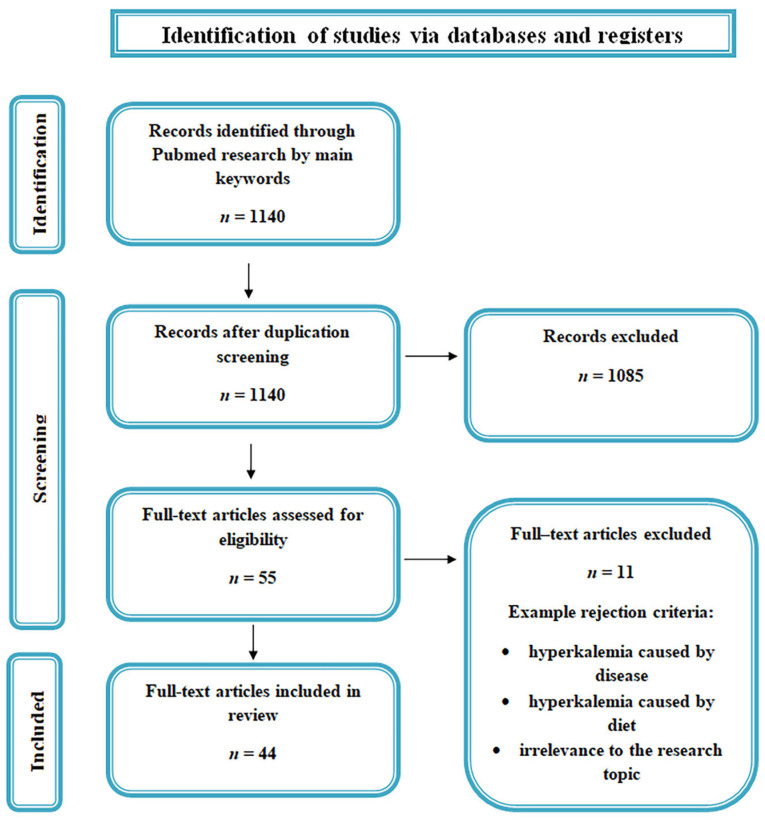
Flow chart of publication selection for the manuscript.

**Table 1 biomedicines-13-00222-t001:** Summary of parameters from selected studies on post-mortem potassium ion concentrations.

Species	Death	Cardiac Alterations	Organ Congestion	Kidney Lesions	Dermal Lesions	Author
Human	Did not occur	Narrow QRS complex, sharp and wide T wave	No data	No data	Cyanosis	Battefort 2012 [16]
Human	Suicide	Elevated K+ concentration in the blood relative to the peripheral blood	No data	No data	Injection mark	Bertol 2012 [26]
Human	Suicide	Did not occur	Liver, spleen, and kidneys (pulmonary edema)	Congestion	Injection mark in the left cubital fossa	Bhatkhande 1977 [22]
human	Did not occur	Bradycardia and hypotonia	No data	Decreased urine volume	No data	Bosse 2011 [21]
Human	Suspicion of suicide	Did not occur	Lungs	No data	Injection marks on the right wrist and the right cubital fossa	Chaturvedi 1986 [8]
Human	Abortion	Salt deposits	No data	Salt deposits	No data	Coulibaly 2010 [10]
Human	Did not occur	Slight tachycardia, pointed T wave, and lower P wave	No data	No data	No data	Iijima 2020 [20]
Human	Did not occur	Wide QRS complex, narrow T wave, left bundle branch block, and asystole	No data	No data	No data	Illingworth 1980 [19]
Human	Did not occur	Bradycardia, hypotonia, and wide QRS complex	No data	No data	No data	John 2011 [17]
Human	Did not occur	Arrhythmia with ventricular tachycardia	No data	No data	No data	Madan 2021 [18]
Human	Did not occur	No data	No data	No data	Third degree chemical burns, nociception disorders, and necrosis	Park 2011 [13]
Human	Autopsy not performed	Myocardial infarction, bradycardia, and asystole	No data	No data	No data	Restuccio 1992 [14]
Human	Did not occur	Pointed T wave and ST elevation	No data	No data	No data	Schaeffer 2018 [25]
Human	Suicide	Cardiac congestion and ventricular dilatation	Heart, lungs, liver, and kidneys	Kidney congestion	Injection marks, slight fresh bleeding	Simon 2023 [3]
Human	Did not occur	Hypotonia, slight tachycardia, wide QRS complex, pointed T wave, and increased blood pressure	No data	No data	No data	Su 2001 [24]
Human	Suicide	Male: Coagulation in the heart Female: Septal hemorrhage	All organs, especially lungs	No data	No data	Watanabe 2011 [9]
Human	Medical mistake	Pointed T wave, wide QRS complex, and ST elevation	No data	No data	No data	Wetherton 2003 [15]
Human	Suicide	Myocardial fibres damage and myocardial interstitial edema	Heart and lungs	No data	Damaged skin structure at injection site	Zhang 2020 [11]
Rabbit	Euthanasia	Dilatation and congestion of the heart and Z-line changes in the heart muscle	Heart	Enlargement of Bowman’s capsule	No data	Zhu 2007 [12]

**Table 2 biomedicines-13-00222-t002:** Summary of methods for detecting deaths due to exogenous hyperkalemia.

Method	Advantages	Disadvantages
Histopathological analysis	Low technical requirements	Non-specificity of pathologies and lack of research
Blood test	Low technical requirements	Low diagnostic value as changes may result from cell lysis and limited post-mortem window
Measuring the concentration of potassium ions in the vitreous body	Potentially high accuracy	Difficulty in sampling, excludes corpses after fires, and limited post-mortem window
Measuring the concentration of potassium ions in the endolymph	Potentially high accuracy	Difficulty in sampling, limited post-mortem window, and lack of research
Aldosterone test	Ease of sampling and resistance against post-mortem changes	Lack of research, excludes individuals with endocrine disorders, and short half-life in blood

## Data Availability

Inquiries can be directed to the corresponding author.

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
