# Peer review of "Detection of Deaths Caused by Hyperkalemia"

_biomedicines, 2025, doi:10.3390/biomedicines13010222_

Round 1
Reviewer 1 Report
Comments and Suggestions for Authors
In this review, the authors focus on the methods used to detect deaths caused by hyperkalemia. Since the identification of deaths caused specifically by hyperkalemia is rather difficult due to their nonspecific symptoms, the authors discuss the use of alternative methods such as histopathological analysis of tissues, assessment of potassium concentration in the vitreous body, and measurement of aldosterone levels. Overall, this review is interesting. However, the authors should further explain the rationale for the alternative methods they suggest, especially regarding aldosterone after death. The authors stated (lines 296-298) “Since it is a steroid hormone, we hypothesize that its concentration may remain relatively stable after death. Therefore, elevated aldosterone levels in postmortem blood could indicate antemortem hyperkalemia by potassium overdose.” Unfortunately, the authors are too affirmative. First, the plasma half-life of aldosterone is not that long, being less than 20 min (1). Moreover, although aldosterone concentration increases during hypokalemia, the pattern of aldosterone protein degradation after death cannot be predicted. In this regard, it is worth mentioning that several nice studies on the role of protein degradation in estimating the postmortem interval and confirming the cause of death in forensic pathology were recently published (2,3), and thus they may be useful to the authors in their current review.
References
1. Awosika, A., Khan, A., Adabanya, U., Omole, A. E., and Millis, R. M. (2023) Aldosterone Synthase Inhibitors and Dietary Interventions: A Combined Novel Approach for Prevention and Treatment of Cardiovascular Disease. Cureus 15, e36184
2. Huang, W., Zhao, S., Liu, H., Pan, M., and Dong, H. (2024) The Role of Protein Degradation in Estimation Postmortem Interval and Confirmation of Cause of Death in Forensic Pathology: A Literature Review. Int J Mol Sci 25
3. Chhikara, A., Kumari, P., Dalal, J., and Kumari, K. (2025) Protein degradation patterns as biomarkers for post-mortem interval estimation: A comprehensive review of proteomic approaches in forensic science. Journal of Proteomics 310, 105326
Author Response
Dear Reviewer 1,
Thank you very much for taking the time to read our manuscript and for your favorable review. 1. In this review, the authors focus on the methods used to detect deaths caused by hyperkalemia. Since the identification of deaths caused specifically by hyperkalemia is rather difficult due to their nonspecific symptoms, the authors discuss the use of alternative methods such as histopathological analysis of tissues, assessment of potassium concentration in the vitreous body, and measurement of aldosterone levels. Overall, this review is interesting. However, the authors should further explain the rationale for the alternative methods they suggest, especially regarding aldosterone after death. The authors stated (lines 296-298) “Since it is a steroid hormone, we hypothesize that its concentration may remain relatively stable after death. Therefore, elevated aldosterone levels in postmortem blood could indicate antemortem hyperkalemia by potassium overdose.” Unfortunately, the authors are too affirmative. First, the plasma half-life of aldosterone is not that long, being less than 20 min (1). Moreover, although aldosterone concentration increases during hypokalemia, the pattern of aldosterone protein degradation after death cannot be predicted. In this regard, it is worth mentioning that several nice studies on the role of protein degradation in estimating the postmortem interval and confirming the cause of death in forensic pathology were recently published (2,3), and thus they may be useful to the authors in their current review. References 1. Awosika, A., Khan, A., Adabanya, U., Omole, A. E., and Millis, R. M. (2023) Aldosterone Synthase Inhibitors and Dietary Interventions: A Combined Novel Approach for Prevention and Treatment of Cardiovascular Disease. Cureus 15, e36184 2. Huang, W., Zhao, S., Liu, H., Pan, M., and Dong, H. (2024) The Role of Protein Degradation in Estimation Postmortem Interval and Confirmation of Cause of Death in Forensic Pathology: A Literature Review. Int J Mol Sci 25
3. Chhikara, A., Kumari, P., Dalal, J., and Kumari, K. (2025) Protein degradation patterns as biomarkers for post-mortem interval estimation: A comprehensive review of proteomic approaches in forensic science. Journal of Proteomics 310, 105326
Answer: Thank you for your positive assessment of our manuscript. After reviewing the literature suggested by the Reviewer regarding aldosterone, we agree that our initial position on this issue was too firm. Considering the aspects mentioned in the review, relevant content was added to the manuscript (in the section 4) and the suggested literature was also cited. We thank you very much for your substantive help.
‘Elevated blood aldosterone levels post-mortem could indicate potassium overdose ante-mortemAlthough the pattern of postmortem change in blood aldosterone concentration has not yet been thoroughly studied, it would be possible to calculate its concentration based on its half-life in the human body (about 20 minutes) [39,40]. Potential advantages of such approach include ease of sampling, resistance to post-mortem concentration changes and ability to calculate the exact concentration of aldosterone before death (based on the time elapsed since death and the half-life). This allows the identification of elevated aldosterone levels secondary to hyperkalemia. Nonetheless, it has limitations, such as reduced reliability in individuals with endocrine disorders for example adrenal dysfunction) and short time to sample due to the short half-life of aldosterone in the blood.‘
Kind regards,
Marta Hałas-Wiśniewska
Reviewer 2 Report
Comments and Suggestions for Authors
Authors reviewed literature to explore a practically efficient way to detect death caused by hyperkalemia. Advantages and disadvantages of various ways are discussed. A couple of approaches are proposed. Overall challenge of specificity has been recognized. The manuscript is well written, but recent advancement in combining ECG with machine learning should be reviewed with emphasis on high sensitivity and specificity. Additionally, several grammar errors need to be corrected.
Lines 41 – 44: deceleration of the slow resting depolarization (what is the “resting depolarization”?), should be the slow component of diastolic depolarization. Action potentials are generated more frequently – fast heart rate, contrary to bradycardia – slow heart rate.
Line 230: missing a word before “hyperkalemia”?
Lines 249-250: provide values of potassium concentration in endolymph.
Line 256: extra “ante-mortem”?
Line 263: missing a word before “hyperkalemia”?
Author Response
Dear Reviewer 2,
Thank you very much for such a detailed review. We took the following comments into account when revising the manuscript and we agree that they increase the value of our work. Below is a list of the changes:
1. The manuscript is well written, but recent advancement in combining ECG with machine learning should be reviewed with emphasis on high sensitivity and specificity. Answer: In the context of the latest reports, the combination of ECG with machine learning is indeed a promising diagnostic method. Most reports definitely concern studies on living patients. However, simulations and pilot studies offer high sensitivity and specificity in detecting various heart diseases, also in the field of forensic medicine, including detection of postmortem hyperkalemia. We thank you for drawing attention to this method, an appropriate comment has been added in the text of the manuscript.
‘In summary, there is a significant lack of research on deaths due to hyperkalemia, leading to a paradox—insufficient research makes it difficult to detect these deaths, and when they go undetected, conducting further research becomes more challenging. We proposed histopathological analysis, measuring the concentration of potassium ions in the vitreous body or endolymph, and blood aldosterone levels as the most promising options. It is also worth mentioning the diagnostic potential of combining electrocardiographic (ECG) analysis with machine learning (ML). Currently, in this context, reports mainly concern studies on living patients. The application of ML technology in the case of death is a challenge due to physiological changes after death, such as tissue breakdown and lack of electrical activity of the heart. However, pilot studies suggest that ML can support the analysis of pathology in the forensic context, e.g. in detecting patterns indicating potassium concentration in tissues in the case of death caused by hyperkalemia. However, further research is necessary to develop procedures adapted to the specificity of postmortem changes [41,42] The methods outlined in the review are not without limitations, each with varying degrees of error. However, analyzing combinations of these parameters may help to ultimately identify hyperkalemia.’
2. Lines 41 – 44: deceleration of the slow resting depolarization (what is the “resting depolarization”?), should be the slow component of diastolic depolarization. Action potentials are generated more frequently – fast heart rate, contrary to bradycardia – slow heart rate. Answer: The context has been corrected.
‘Moreover, hyperkalemia significantly affects the circulatory system. By reducing the threshold potential, it increases the cell membrane’s conductivity to potassium ions in cardiomyocytes, leading to a deceleration of the slow resting depolarization. As a result, action potentials are generated less frequently in the pacemaker cells of the heart, manifesting as pointed T waves, widened QRS complexes, bradycardia, and even asystole. In the later stages, this condition can progress to ventricular fibrillation and ultimately, death [1,3–6].
3. Line 230: missing a word before “hyperkalemia”? Answer: The word was unnecessary in this context and has been deleted.
4. Lines 249-250: provide values of potassium concentration in endolymph. Answer: The value was added based on literature. ‘Under physiological conditions, the endolymph contains a significantly higher potassium concentration compared to the cytoplasm of surrounding epithelial cells. In the article from 2006 the author assesses the potassium concentration for the endolymph as 157mM in comparison to the 5mM of blood plasma. This elevated potassium level distinguishes endolymph from other extracellular fluids [34]’
5. Line 256: extra “ante-mortem”?
Answer: The extra word has been removed
6. Line 263: missing a word before “hyperkalemia”? Answer: The word was unnecessary in this context and has been deleted.
Kind regards,
Marta Hałas-Wiśniewska
Round 2
Reviewer 1 Report
Comments and Suggestions for Authors Although the authors did not fully address my concerns, the article is acceptable in its current version.
Reviewer 2 Report
Comments and Suggestions for Authors
My concerns in the original manuscript have been addressed in this revision.